# In Vivo and In Vitro Evidence for an Interplay between the Glucocorticoid Receptor and the Vitamin D Receptor Signaling

**DOI:** 10.3390/cells12182291

**Published:** 2023-09-15

**Authors:** Maud Bagnoud, Jana Remlinger, Marine Massy, Dmitri Lodygin, Anke Salmen, Andrew Chan, Fred Lühder, Robert Hoepner

**Affiliations:** 1Department of Neurology, Inselspital, Bern University Hospital, 3010 Bern, Switzerland; jana.remlinger@unibe.ch (J.R.); marine.massy@insel.ch (M.M.); anke.salmen@insel.ch or anke.salmen@rub.de (A.S.); andrew.chan@insel.ch (A.C.); robert.hoepner@insel.ch (R.H.); 2Department of Biomedical Research, University of Bern, 3010 Bern, Switzerland; 3Graduate School for Cellular and Biomedical Sciences, University of Bern, 3012 Bern, Switzerland; 4Institute for Neuroimmunology and Multiple Sclerosis Research, University Medical Center Göttingen, 37075 Göttingen, Germany; dmitri.lodygin@med.uni-goettingen.de (D.L.); fred.luehder@med.uni-goettingen.de (F.L.)

**Keywords:** experimental autoimmune encephalomyelitis, glucocorticoid, vitamin D

## Abstract

Our previous work demonstrated that vitamin D (VitD) reduces experimental autoimmune encephalomyelitis (EAE) disease severity in wild-type (WT) but not in T cell-specific glucocorticoid (GC) receptor (GR)-deficient (GR^lck^) mice. This study aimed to investigate the interplay between the GR- and VitD receptor (VDR) signaling. In vivo, we confirmed the involvement of the GR in the VitD-induced effects in EAE using WT and GR^lck^ mice. Furthermore, we observed that VitD-enhanced T cell apoptosis and T regulatory cell differentiation are diminished in vitro in CD3+ T cells of GR^lck^ but not WT mice. Mechanistically, VitD does not appear to signal directly via the GR, as it does not bind to the GR, does not induce its nuclear translocation, and does not modulate the expression of two GR-induced genes. However, we observed that VitD enhances VDR protein expression in CD3+ T cells from WT but not GR^lck^ mice in vitro, that the GR and the VDR spatially co-localize after VitD treatment, and that VitD does not modulate the expression of two VDR-induced genes in the absence of the GR. Our data suggest that a functional GR, specifically in T cells, is required for the VDR to signal appropriately to mediate the therapeutic effects of VitD.

## 1. Introduction

Glucocorticoids (GCs) are lipid-soluble steroid hormones that diffuse passively through the cell membrane and mainly induce effects via their binding to the glucocorticoid (GC) receptor (GR), a nuclear receptor [1]. In the absence of its ligand, the GR mostly localizes in the cytoplasm in a multi-protein complex that prevents degradation and favors its maturation [2]. Once bound to GCs, the GR translocates into the nucleus, dimerizes, binds on GC response elements (GREs), and induces or represses the expression of a plethora of genes, such as *Dusp1* and *Tsc22d3*, important in controlling inflammation [3,4]. Due to their potent immunosuppressive and anti-inflammatory properties, GCs exert beneficial clinical effects in a large number of human diseases, such as multiple sclerosis (MS), and their animal models, such as experimental autoimmune encephalomyelitis (EAE) [5]. Here, GCs have been shown to reduce the expression levels of pro-inflammatory cytokines, chemokines, and adhesion molecules [5], to modulate the migration of activated T cells [6], to increase spinal cord T cell apoptosis [7], and to enhance IL-2 dependent expansion of Foxp3+ Treg cells [8]. In T cell-specific GR-deficient mice, the therapeutic effects of GCs are absent, indicating a crucial role of the peripheral T cells in mediating GC effects [9].

Vitamin D (VitD) is a lipid-soluble vitamin acting as a hormone [10,11]. It exists in two chemically slightly different forms comprising its mainly metabolically inactive storage form calcidiol (25-OH vitamin D3), and its metabolically active form calcitriol (1,25-(OH)2 vitamin D3, (1,25D)), predominantly mediating clinical effects [12,13]. The binding of calcitriol to its nuclear receptor, the VitD receptor (VDR), leads to heterodimerization with the unbound retinoid X receptor (RXR). This cytoplasmic complex then translocates into the nucleus, binds to VitD response elements (VDREs), and modulates the expression of various genes [14]. VitD is well recognized for its immuno-modulatory properties [13,15]. It has been shown to modulate the composition of T helper cell subsets by inhibiting Th1 and Th17 polarization [16,17] and favoring Th2 cell development [17], as well as enhancing the differentiation of T regulatory (Treg) cells [18,19]. In vivo, calcitriol-treated EAE-diseased mice exhibit a lower incidence of the disease, a decreased peak disease severity, and a reduced cumulative disease index [20]. Calcitriol-induced responses, such as reduced spinal cord white matter demyelination or decreased spinal cord immune cell infiltration, are already observable within the first three days after starting the treatment [21]. When given in a preventive setting, calcitriol (50 ng/day) reduces EAE disease severity in WT mice but not in mice with a T cell-specific VDR deficiency, suggesting that the VDR is a direct target of the therapeutic effects of calcitriol [22]. However, the regulation of the GR in T cell-specific VDR-deficient mice was not investigated in this study. Our previous work demonstrated a calcitriol-dependent upregulation of the GR expression in T cells, increasing the GC signaling axis with the known downstream immunological effects [23]. We also observed that calcitriol at a concentration of 10 ng/day given over three consecutive days after the appearance of the first clinical symptoms reduces EAE disease severity in WT but not in T cell-specific GR-deficient mice, suggesting a potential involvement of the GR in VitD signaling. Thus, a deficient GR regulation must be considered in T cell-specific VDR-deficient mice due to decreased VitD signaling. This highlights the need to detail the efficacy of VitD treatment in more depth, considering both receptors.

In the present study, we investigated if calcitriol potentially signals directly via the GR in T cells [24] or if the GR, specifically in T cells, is necessary for the VDR to signal appropriately in order to mediate the therapeutic effects of calcitriol.

## 2. Materials and Methods

Animal studies were approved by the local authorities (Office of Agriculture and Nature Bern, Switzerland: 25/19 and 32/22). GR^wt^ and GR^lck^ (Nr3c1^tm2GSc^Tg^(lck-cre)1Cwi^) mice on a C57BL/6 background were housed under conventional housing conditions at the in-house animal facility of the University of Bern. GR^lck^ mice lack GR expression, specifically in T cells [25].

Experimental autoimmune encephalomyelitis: Active MOG_35–55_ EAE was induced in 8–12 weeks-old female GR^wt^ and GR^lck^ mice following our previously described protocol [23,26]. Briefly, animals were immunized by subcutaneous injection of 100 µg MOG_35–55_ peptide (Institute of Medical Immunology, Charité, Berlin, Germany) in PBS (VWR, PA, USA) emulsified in complete Freund’s adjuvant containing 100 µg mycobacterium tuberculosis (Difco, MA, USA), followed by 200 ng pertussis toxin (Quadratech, Eastbourne, UK, intraperitoneal (i.p.) injection, days 0 and 2 after immunization). Animals were scored daily in a blinded manner using a 10-point EAE scale. Treatment was initiated when animals had an EAE score ≥ 2. Calcitriol (1-, 10-, 100-, or 1000 ng/day (Selleckchem, TX, USA); DMSO (Merck, Darmstadt, Germany) in peanut oil (Migros); oral gavage, or control (DMSO (Merck, Darmstadt, Germany) in peanut oil (Migros); oral gavage, were given on three consecutive days.

Apoptosis and Treg differentiation assays: CD3+ T cells were isolated from the spleen of GR^wt^ and GR^lck^ mice by negative magnetic-activated cell sorting (MACS) selection (Pan-T cell isolation kit^®^, Miltenyi Biotec, Bergisch-Gladbach, Germany). They were resuspended at 1 mio/mL in T cell medium (RPMI 1640 medium, Thermo Fisher Scientific, MA, USA) containing 10% fetal bovine serum (FBS, Thermo Fisher Scientific, MA, USA), 1% Penicillin/Streptomycin (P/S, Thermo Fisher Scientific, MA, USA), and 1% L-glutamine (Thermo Fisher Scientific, MA, USA). Cells were stimulated with concanavalin A (conA, 1.5 μg/mL, Sigma-Aldrich, MO, USA), treated with calcitriol 10- or 1000 nM (Selleckchem, TX, USA), and plated in a round bottom 96-well plate for 24 h at 37 °C. Apoptosis (Annexin V/Propidium iodide, Becton Dickinson Bioscience (BD), NJ, USA) and Treg differentiation (Treg Detection Kit (CD4/CD25/FoxP3), Miltenyi Biotec, Bergisch-Gladbach, Germany) were analyzed by flow cytometry (FACS SORP LSRII, BD, NJ, USA) following the manufacturer’s protocols.

GR binding assay: A LanthaScreen time-resolved fluorescence resonance energy transfer (TR-FRET) competitive binding assay was performed by Thermo Fisher Scientific (MA, USA) according to their protocol. A human recombinant GR and different concentrations of calcidiol, calcitriol, methylprednisolone (MP), and dexamethasone (DEX) (for each: 0.15–3000 nM; 3-fold dilutions; 10 points in duplicates) were used.

GR nuclear translocation: HeLa ω3 clone stably expressing the mCherry-H2B-T2A-NFAT_1–410_-EYFP cassette under the control of the CAG promoter was previously generated as a reporter cell line for Ca^2+^-dependent NFAT translocation [27]. In this cell line, red fluorescence (mCherry) is confined to the nuclei. To eliminate the EYFP fluorescence resulting in HeLa cells only expressing mCherry in the nucleus, we transiently transfected HeLa ω3 cells with a CRISPR/Cas9-expressing plasmid (pX330, Addgene, MA, USA) and a U6 promoter-driven short guide RNA construct targeting the coding sequence of EYFP (GAAGTTCGAGGGCGACACCCTGG) using Lipofectamine 3000 (Thermo Fisher Scientific, MA, USA). One week after transfection, mCherry-H2B+ HeLa ω3 cells that had lost EYFP expression were isolated by flow cytometry (FACS Aria II SORP, BD, NJ, USA) and expanded in culture. The proper nuclear localization of mCherry was verified in the edited cell line by fluorescence microscopy. Hela ω3 cells expressing mCherry-H2B in the nucleus were cultured in DMEM (Gibco, MT, USA) containing 1% L-glutamine 200 mM (Biowest, Pays de la Loire, France), 1% P/S (Gibco, MT, USA), 1% Na-Pyruvat 100 mM (Biowest, Pays de la Loire, France), 1% nonessential amino acids 100X (Gibco, MT, USA), 1% L-asparagine monohydrate (Sigma-Aldrich, MO, USA), 0.0004% 2-mercaptoethanol (Sigma-Aldrich, MO, USA), and 10% GC-stripped FBS (Sigma-Aldrich, MO, USA) until confluency. For the experiment, cells were incubated for 2 h with MP 100 nM (Pfizer, NY, USA) as positive control or different concentrations of calcitriol (40 nM, 400 nM, 4 μM, or 100 μM, Selleckchem, TX, USA). Afterward, cells were harvested using 2 mM EDTA in PBS (VWR, PA, USA), fixed with 2 mL BD Phosphoflow Lyse/Fix buffer (BD, NJ, USA) at 37 °C for 10 min, and incubated with 1 mL BD Phosphoflow perm buffer III (BD, NJ, USA) for 30 min at −20 °C. Staining was performed for 20 min at RT with a monoclonal rabbit anti-GR (Glucocorticoid Receptor (D8H2) XP Rabbit mAb, Cell signaling, MA, USA) and, subsequently, anti-rabbit IgG F(ab′)2 fragment (Alexa Fluor 488 conjugate, Cell signaling, MA, USA) [28]. A total of 500–1000 cells per sample were analyzed using an ImageStream system (Luminex). Data are presented as normalized similarity index between the mCherry-labeled nucleus and the AF488 staining of the GR inside the cell.

*GR*- and *VDR*-induced gene expression (Real-time quantitative polymerase chain reaction (RT-qPCR)): CD3+ T cells were isolated from the spleen of GR^wt^ and GR^lck^ mice and resuspended in T cell medium as described in previous sections. Cells were stimulated with conA (1.5 μg/mL, Sigma-Aldrich, MO, USA), treated with MP 60 nM (Pfizer, NY, USA) or calcitriol 10 nM (Selleckchem, TX, USA), and plated in a flat bottom 24-well plate (Sarstedt, Nümbrecht, Germany) for 3 h at 37 °C. The expression of *Dusp1* (forward sequence: CAACCACAAGGCAGACATCAGC, reverse sequence: GTAAGCAAGGCAGATGGTGGCT) and *Tsc22d3* (forward sequence: TCAATGAGGGCATCTGCAACCG, reverse sequence: CATCAGGTGGTTCTTCACGAGG) genes was analyzed in GR^wt^ mice and the expression of *Cdkn1b* (forward sequence: AGCAGTGTCCAGGGATGAGGAA, reverse sequence: TTCTTGGGCGTCTGCTCCACAG) and *Ninj1* (forward sequence: GTGGTCCTCATCTCTATCTCCC, reverse sequence: CGACGATGATGAAAACCAGTCCC) genes in GR^wt^ and GR^lck^ mice by RT-qPCR using the SYBR Green technology (LightCycler^®^ 480 SYBR Green I Master, Roche, Basel, Switzerland). *Actb* (forward sequence: CATTGCTGACAGGATGCAGAAGG, reverse sequence: TGCTGGAAGGTGGACAGTGAGG) was used as a housekeeping gene. Experiments were run on a Light Cycler 480 Roche machine. The relative gene expression was calculated using the comparative quantification method.

GR-VDR co-localization and expression: CD3+ T cells were isolated from the spleen of GR^wt^ mice for the co-localization analysis and of GR^wt^ and GR^lck^ mice for the expression analysis and resuspended in T cell medium as described in previous sections. Cells were stimulated with conA (1.5 μg/mL, Sigma-Aldrich, MO, USA), treated with MP 60 nM (Pfizer, NY, USA) or calcitriol 10 nM (Selleckchem, TX, USA), and plated in a flat bottom 24-well plate (Sarstedt, Nümbrecht, Germany) with wells containing a cover glass coated with 0.1 mg/mL of Poly-L-lysin (Merck, Darmstadt, Germany). After 2 h at 37 °C, cells were washed with PBS + 0.1% Tween 20 (PBST; PBS: VWR, PA, USA; Tween 20: AxonLab, Baden, Switzerland) and fixed with 4% paraformaldehyde (PFA, Grogg Chemie, Stettlen-Deisswil, Switzerland) for 10 min at 4 °C. Cells were then washed three times with PBST, and Image-iT Fx signal enhancer (Invitrogen, Massachusetts, USA) was applied for 30 min at RT. Blocking was performed with 5% bovine serum albumin (BSA, Lee BioSolutions, MO, USA) + 0.05% Tween 20 (AxonLab, Baden, Switzerland) in PBS (VWR, Pennsylvania, USA) for 30 min at RT. Cells were then incubated ON at 4 °C with anti-GR (Glucocorticoid Receptor (D8H2) XP Rabbit mAb, Cell signaling, MA, USA) and anti-VDR (VDR Monoclonal Antibody (9A7) (Rat/IgG2b), (Thermo Fisher Scientific, Massachusetts, USA) antibodies diluted in PBS (VWR, PA, USA) + 10 mM glycine (Thermo Fisher Scientific, Massachusetts, USA) + 0.05% Tween 20 (AxonLab, Baden, Switzerland) + 0.1% Triton X100 (Sigma-Aldrich, MO, USA) + 0.1% H_2_O_2_ (Bern hospital pharmacy) [29]. After three steps of washing with PBST, cells were incubated for 2 h at RT with the secondary antibodies (GR: Goat-α-Rabbit Alexa Fluor^®^ 555 (2 mg/mL); VDR: Goat-(ab′)2 fragment α-Rat Alexa Fluor^®^ 488). Cells were rewashed three times with PBST, and the fluo-stained cover glasses were mounted on slides with DAPI (BioConcept, Basel, Switzerland). Three images per condition were acquired with a confocal microscope (Olympus fluoview FV1000). Co-localization and expression (intensity) of both receptors were analyzed with ImageJ (National Institute of Mental Health, NIH, Bethesda, USA, version 1.52p).

Statistics: Data are expressed as mean +/− standard error of the mean (SEM). Comparison of data was performed using the Kruskal–Wallis Test (KWT) for independent, and Wilcoxon signed rank sum test (WSRST) or Friedman analysis of variance (ANOVA) test (FT), depending on the number of group comparisons, for dependent measurements. Statistics from the GR nuclear translocation assay were performed using a Repeated Measure One-way ANOVA followed by the Newman Keuls Multiple Comparison test (RMOW-ANOVA). The *p*-value was considered significant when below 0.05.

## 3. Results

### 3.1. Calcitriol Improves MOG_35–55_ EAE Disease Course in WT but Not in GR^lck^ Mice In Vivo

To verify the importance of the GR for the in vivo effects of calcitriol, MOG_35–55_ EAE was induced in both WT and GR^lck^ mice, and mice were treated over three consecutive days with different concentrations of calcitriol. The three lowest concentrations of calcitriol significantly reduced the mean cumulative MOG_35–55_ EAE score compared to the control WT mice (calcitriol 1 ng: KWT: *p* < 0.05; calcitriol 10 ng: KWT: *p* < 0.0001; calcitriol 100 ng: KWT: *p* < 0.0001; calcitriol 1 µg: KWT: *p* > 0.05; Figure 1). In contrast, calcitriol treatment did not improve the mean cumulative MOG_35–55_ EAE score as compared to the control group in GR^lck^ mice. Indeed, calcitriol 10 ng and 1 µg worsened the disease (calcitriol 1 ng: KWT: *p* > 0.05; calcitriol 10 ng: KWT: *p* < 0.0001; calcitriol 100 ng: KWT: *p* > 0.05; calcitriol 1 µg: KWT: *p* < 0.0001; Figure 1).

### 3.2. Calcitriol Induces Treg Differentiation and Increases the Apoptosis of CD3+ T Cells from WT but Not GR^lck^ Mice In Vitro

To investigate the importance of the GR for the calcitriol-induced effects in vitro, CD3+ T cells of both WT and GR^lck^ mice were treated with calcitriol, and Treg differentiation, as well as apoptosis, were assessed. Treatment of CD3+ T cells of WT mice with calcitriol 1 µM resulted in an increased percentage of Foxp3+ cells in CD4 + CD25 high T cells and enhanced apoptosis of approximately 2% than controls ((A, B, GR^wt^): FT: *p* < 0.05; Figure 2). In contrast, no significant changes were observed in CD3+ T cells of GR^lck^ mice ((A, B, GR^lck^): FT: *p* > 0.05; Figure 2).

### 3.3. Neither Calcitriol nor Calcidiol Bind to a Human Recombinant GR

Because the first experiments suggested that the presence of the GR impacts the VitD effects on T cells, we wanted to evaluate whether calcitriol is able to directly bind to the GR. We observed that in contrast to the positive controls DEX and MP that displaced 100% of the tracer, neither calcidiol nor calcitriol was able to remove it from the binding site of the GR (Figure 3), suggesting that direct binding of both calcidiol and calcitriol to the GR is negligible.

### 3.4. Calcitriol Does Not Induce the Nuclear Translocation of the GR In Vitro

To further corroborate these findings, we wanted to determine if calcitriol induces the nuclear translocation of the GR. Data are shown as the normalized similarity between the nucleus and the GR inside the cell and demonstrated that in contrast to MP 100 nM (RMOW-ANOVA: *p* < 0.001; Figure 4), all the tested concentrations of calcitriol did not induce the translocation of the GR into the nucleus (RMOW-ANOVA: *p* > 0.05; Figure 4). Representative images of the staining are presented in Appendix A.

### 3.5. Calcitriol Does Not Modulate the Expression of Two GR-Induced Genes, Dusp1 and Tsc22d3, in CD3+ T Cells from WT Mice In Vitro

To further investigate the involvement of the GR in calcitriol signaling, the expression of two GR-induced genes, *Dusp1* and *Tsc22d3,* was analyzed in vitro in CD3+ T cells of WT mice. In contrast to MP 60 nM, which significantly increased the expression of *Dusp1* and *Tsc23d3* as compared to untreated cells (FT: *p* < 0.0001; Figure 5), calcitriol 10 nM did not modulate the expression of both genes (FT: *p* > 0.05; Figure 5).

The results from all these experiments strongly argue against the direct use of the GR for VitD signaling: calcitriol does not directly bind to the GR, nor does it support nuclear translocation of the GR nor modulation of genes downstream of known GR signaling cascades.

### 3.6. Calcitriol Enhances VDR Protein Expression in CD3+ T Cells from WT but Not GR^lck^ Mice In Vitro

The expression of the VDR and GR was analyzed in vitro in CD3+ T cells of WT and GR^lck^ mice and in CD3+ T cells of WT mice, respectively. Both treatments significantly enhanced the expression of the VDR as compared to controls in CD3+ T cells of WT mice ((A, GR^wt^): MP 60 nM: KWT: *p* < 0.001, calcitriol 10 nM: KWT: *p* < 0.0001; Figure 6) but not in CD3+ T cells of GR^lck^ mice ((A, GR^lck^): MP 60 nM: KWT: *p* > 0.05, calcitriol 10 nM: KWT: *p* > 0.05; Figure 6). Both treatments also increased the expression of the GR in CD3+ T cells of WT mice as compared to controls ((B, GR^wt^): MP 60 nM: KWT: *p* < 0.0001, calcitriol 10 nM: KWT: *p* < 0.0001; Figure 6). Representative images of the staining are presented in Appendix A.

### 3.7. Calcitriol Induces the Nuclear Co-Localization of the GR and VDR in CD3+ T Cells from WT Mice In Vitro

To better understand the role of the GR in calcitriol signaling, the localization of the GR and VDR was analyzed in vitro in CD3+ T cells of WT mice. Both treatments significantly increased the nuclear co-localization of the GR and VDR as compared to controls when the percentage of overlapping stained pixels was normalized to the VDR or GR area ((A): MP 60 nM: KWT: *p* < 0.01, calcitriol 10 nM: KWT: *p* < 0.05; (B): MP 60 nM: KWT: *p* < 0.01, calcitriol 10 nM: KWT: *p* < 0.0001; Figure 7). Representative images of the staining are presented in Appendix A.

### 3.8. Calcitriol Enhances the Expression of Two VDR-Induced Genes, Cdkn1b and Ninj1, in CD3+ T Cells from WT but Not GR^lck^ Mice In Vitro

The expression of the two VDR-induced genes, *Cdkn1b* and *Ninj1,* was analyzed in vitro in CD3+ T cells of WT and GR^lck^ mice treated with calcitriol. Calcitriol 10 nM significantly increased the gene expression of *Cdkn1b* and *Ninj1* in CD3+ T cells of WT mice as compared to untreated cells (*Cdkn1b*: WSRST: *p* < 0.01; *Ninj1*: WSRST: *p* < 0.01; Figure 8). In contrast, no significant changes in *Cdkn1b* and *Ninj1* expression were observed in CD3+ T cells after calcitriol treatment when the GR was absent (*Cdkn1b*: WSRST: *p* > 0.05; *Ninj1*: WSRST: *p* > 0.05: Figure 8).

## 4. Discussion

The receptors for GCs and VitD are steroid receptors of the nuclear receptor family located in the cytoplasm. They have similar modes of action [30,31] and share structural similarities [10,11], such as their three major domains [32]. There are, however, only very limited data about a possible interaction and a functional interplay between them. We, therefore, asked whether intracellular signaling pathways of one of these two receptors are influenced by concurrent activation of the other. In the first step, we wanted to verify the importance of the GR, specifically in T cells, for the immunological effects of calcitriol in an in vivo EAE model. We clearly demonstrated that the beneficial effects of calcitriol (1–100 ng/day) on EAE observed in vivo in WT mice were lost in T cell-specific GR-deficient mice. In the early 1990s, i.p. injection of calcitriol every other day for 15 days, starting three days before EAE induction, was shown to significantly suppress EAE severity [33]. It was also shown that calcitriol supplementation abrogates EAE onset when given in a preventive setting [34]. Clinically, calcitriol treatment has been demonstrated to decrease the paralysis of the mice 48 h post-treatment [21], supporting the rapid effects of calcitriol that we observed in our experiments. In contrast, our results did not show any beneficial effects of calcitriol on EAE disease course in T cell-specific GR-deficient mice, supporting the relevance of a functional GR for proper calcitriol-induced effects. However, experimental data demonstrated that a functional VDR is needed to mediate calcitriol-induced effects as well. Calcitriol-treated VDR-deficient EAE-diseased mice do not show any decreased EAE incidence, delayed onset, decreased peak severity, or reduced cumulative disease index in contrast to calcitriol-treated WT EAE-diseased mice [22]. These findings suggest that both a functional GR and a functional VDR are required to mediate calcitriol-induced effects in the EAE, further hinting at crosstalk between both GC and VitD signaling pathways. The worsening of EAE in the highest calcitriol concentration group (1 µg/day) occurring in both genotypes may be mediated by different mechanisms. Detrimental effects of ultra-high concentrations of cholecalciferol, a precursor of calcitriol, on EAE severity were reported [35]. As an underlying mechanism, they observed hypercalcemia caused by high but not medium or low doses of cholecalciferol, presumably unrelated to the GR. They deduced that it could be responsible for the different observed side effects of cholecalciferol [35].

Corroborating our in vivo results, we demonstrated that calcitriol induced T cell apoptosis and enhanced Treg differentiation in vitro in CD3+ T cells of WT mice but that these effects were completely gone when the T cells were deficient for the GR. This further shows that a functioning GR is mandatory for at least some effects mediated by the VDR signaling pathway. Calcitriol can produce a tolerogenic environment via the induction of Treg cells. Human CD4+ T cells can convert calcidiol into calcitriol, leading to an increased expression of FOXP3, CD25, and CTLA-4 [36]. Calcitriol has been shown to induce Treg cells from stimulated murine CD4+ T cells in vitro [37]. In vivo, calcitriol increases the mRNA expression of *FoxP3* in splenocytes of EAE-diseased mice [38]. The effect of calcitriol on apoptosis is less clear, as the sensitivity of the cells to calcitriol-induced apoptosis appears to be dependent on the health status of the cells. Calcitriol modulates apoptosis in different ways, notably by enhancing the cell death of abnormal cells and by decreasing the apoptosis of healthy cells. For instance, calcitriol strongly affects the viability of several cancer cell lines, whereas calcitriol protects β-cells against high glucose-induced apoptosis [39,40,41]. In the context of EAE, calcitriol has been shown to enhance the sensitivity of pathogenic T cells to apoptosis in vivo by increasing the level of pro-apoptotic genes and decreasing the level of anti-apoptotic genes [42,43]. In contrast, no altered cell apoptosis was demonstrated after calcitriol treatment of stimulated splenic-derived T cells of naïve cholecalciferol-treated WT mice [35].

The question arises as to what the molecular basis of the dependence of effective VDR signaling on the concomitant presence and function of the GR could be. A trivial explanation would be that calcitriol potentially signals in T cells via the GR itself. Alternatively, it can be envisioned that the GR, specifically in T cells, is important for effective VDR signaling by an interaction with the VDR and, therefore, is required for mediating the therapeutic effects of calcitriol. To address the first hypothesis, we evaluated whether calcitriol is able to influence different steps in the signaling pathway of GCs. For this, we determined if calcitriol is able to (i) directly bind to the GR, (ii) induce the nuclear translocation of the GR, and (iii) modulate the expression of two well-recognized GR-induced genes. The ability of calcitriol to bind to the GR was assessed through a competitive assay using the TR-FRET technology. A human recombinant GR was used in this assay, and different concentrations of calcidiol, calcitriol, and GCs were tested. We observed that in contrast to DEX and MP, calcidiol and calcitriol do not bind to the human GR. These results contradicted findings demonstrating that both calcidiol and calcitriol bind to the GR even at a greater affinity than DEX and MP [24]. To confirm our results, we showed that in contrast to MP, calcitriol does not induce the nuclear translocation of the GR in the reporter HeLa ω3 cell line and does not significantly modulate the expression of the two GR-induced genes *Dusp1* and *Tsc22d3* in splenic-derived CD3+ T cells of WT mice. In addition, we searched a public database (GSE154741 from [44]), where human CD4+ memory T cells were treated with VitD and sequenced 48 h afterwards. From the overlap of 169 genes of a list of GR-responsive genes [45], only 4 (2.4%) were upregulated more than 1.5 fold after VitD treatment (*Abca1, Pygl, Nab1, S100a13*), arguing that the upregulation of GR-responsive genes by VitD is rather a rare event and that VitD is not able to upregulate GR-responsive genes in a systematic way. From these results, we concluded that calcitriol does not appear to signal through the GR.

To evaluate our second hypothesis, we first analyzed the effect of calcitriol and MP treatments on the protein expression of the VDR and GR in CD3+ T cells from WT and T cell-specific GR-deficient mice. It was observed that both treatments significantly increased the VDR protein expression in CD3+ T cells from WT but not T cell-specific GR-deficient mice, suggesting a role of the GR in calcitriol signaling. Furthermore, calcitriol and MP significantly enhanced the expression of the GR in CD3+ T cells from WT mice, corroborating previous data from our lab [23]. Next, we investigated if calcitriol treatment leads to the co-localization of the GR and VDR. To this end, we chose an object-based approach, which measures the spatial relationship between distinct objects and gives us a quantification of co-localization [46]. Our data demonstrated that calcitriol treatment enhances the co-localization of both receptors in the nucleus of CD3+ T cells of WT mice, as shown by an increased percentage of overlap between the GR and the VDR staining. These results suggest that there is a spatial overlap of the two receptors. The GR and the VDR have already been shown to interact via the transcriptional co-activator mediator complex subunit 14 (MED14) in human peripheral blood mononuclear cells [47]. MED14 was shown to link the ligand-bound VDR and GR at the adjacent VDRE and GRE of the human MKP-1 gene promoter. Last, we examined whether calcitriol is able to induce VDR-responsive gene expression in T cells in the absence of the GR. We could show that two calcitriol-induced genes, *Cdkn1b* and *Ninj1*, are upregulated after calcitriol treatment in WT but not in T cell-specific GR-deficient mice, further supporting a possible interaction between the GR and the VDR.

Altogether, our results suggest an interplay between GC and VitD signaling, supported by several lines of evidence. First, stimulation of any one of the GR or VDR positively influences the expression levels of the respective other receptor. GC treatment has been shown to upregulate the VDR [48,49], and we demonstrated an enhanced GR expression in human CD3+ T cells after calcitriol treatment [23]. In vivo, VitD supplementation has been shown to modulate the impaired GR signaling pathway induced by a long-term administration of GCs in bone marrow cells [50]. Ex vivo, calcitriol increases the sensitivity to GCs of activated human blood-derived Th17.1 cells, known to be poor GC responders, by further reducing cell proliferation and by downregulating the expression of pro-inflammatory and brain-homing markers [51]. Second, both steroids are known to influence the efficacy of each other. We previously showed that calcitriol increases GC efficacy in the context of acute MS relapses [23]. Others observed enhanced VitD efficacy by GCs in an animal model of Crohn’s disease [52]. Mechanistic data also support these functional findings [47]. Furthermore, it was observed that DEX enhances the mouse VDR protein expression in a murine squamous cell carcinoma model by increasing VDR transcription in a GR-dependent manner. The binding of the GR to a specific GRE upstream of the VDR promoter also modulates the expression of several VDR target genes [53]. In this study, we extended these findings by showing that effective VitD signaling only occurs in T cells in the presence of the GR in vitro and in vivo and by highlighting a co-localization of the GR and VDR in the nucleus after stimulation. This strongly argues for specific interactions between GR and VDR signaling, the molecular mechanism underlying this relationship being the target of future studies.

Our study investigated the relevance of the GR, specifically in T cells, for proper calcitriol signaling through the VDR. However, it could be pertinent to determine if the VDR plays an important role in GC signaling as it might have implications for GC-based MS relapse treatment. Also, our data are mainly based on animal samples, making experiments with human samples mandatory to estimate if findings can be translated back to the bedside.

## 5. Conclusions

The findings of our experimental study demonstrated that a complex interplay between GR- and VDR signaling exists. We could show that calcitriol does not appear to signal via the GR, but we assume that the GR is required for the VDR to signal properly, specifically in CD3+ T cells, to mediate the therapeutic effects of calcitriol. This should promote further investigations as both GCs and VitD are common treatments, and a better understanding of their signaling with a broader scope to generalize our preliminary findings would have clinical implications.

## Figures and Tables

**Figure 1 cells-12-02291-f001:**
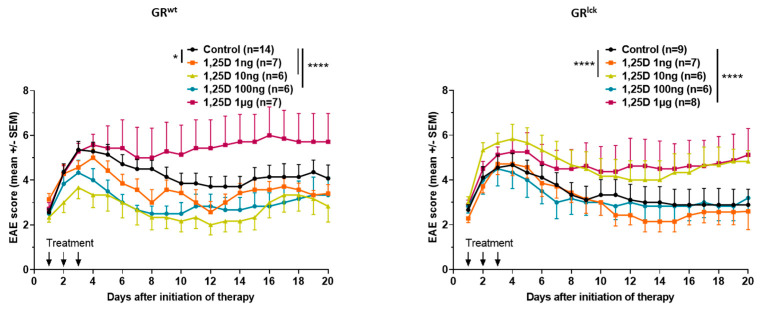
Clinical disease course of MOG_35–55_ EAE in WT and GR^lck^ mice treated for three consecutive days with control (DMSO in peanut oil) or different concentrations of calcitriol (1 ng, 10 ng, 100 ng, and 1 µg). Number of included animals are displayed in the graph. EAE score: 10 score system. Abbreviations: EAE: experimental autoimmune encephalomyelitis, GR^lck^: T cell-specific GR-deficient mice, GR^wt^: WT mice, SEM: standard error of the mean, 1,25D: calcitriol. Statistics: Kruskal–Wallis test: * < 0.05, **** < 0.0001.

**Figure 2 cells-12-02291-f002:**
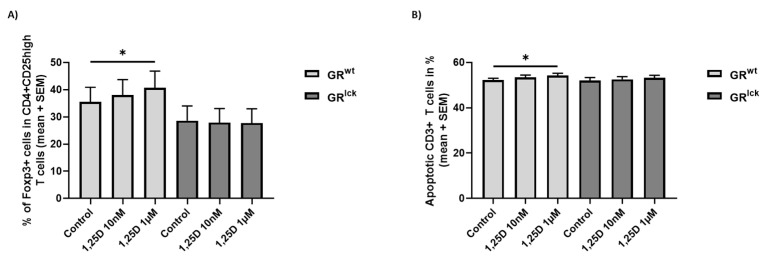
(**A**) Treg differentiation and (**B**) apoptosis of stimulated splenic-derived CD3+ T cells of WT and GR^lck^ mice ((**A**): GR^wt^: *n* = 5 in 1 to 3 replicates, GR^lck^: *n* = 5 in triplicates; (**B**): GR^wt^: *n* = 5 in triplicates, GR^lck^: *n* = 5 in triplicates; stimulus conA 1.5 µg/mL; 24 h). Incubation with control, calcitriol 10 nM or calcitriol 1 µM ((**A**): Treg kit, (**B**): Annexin V/PI, flow cytometry). Abbreviations: GR^lck^: T cell-specific GR-deficient mice, GR^wt^: WT mice, SEM: standard error of the mean, 1,25D: calcitriol. Statistics: Friedman ANOVA test: * < 0.05.

**Figure 3 cells-12-02291-f003:**
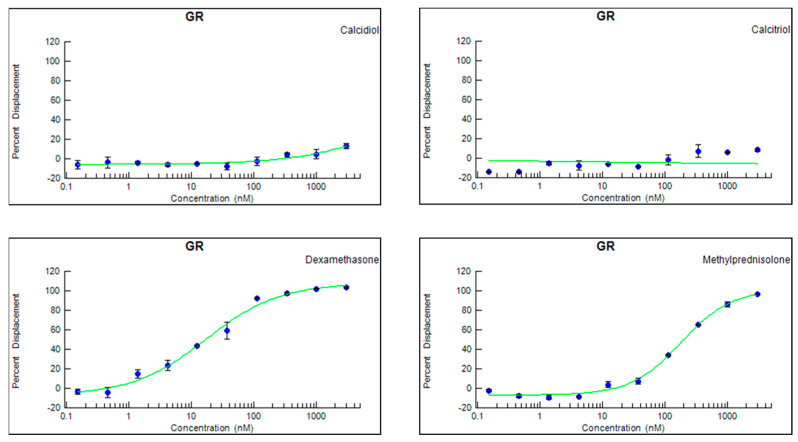
The ability of calcidiol, calcitriol, dexamethasone, and methylprednisolone to bind to a human recombinant GR was checked through a competitive binding assay using the time-resolved fluorescence resonance energy transfer technology (experiment in duplicates (0.15–3000 nM, 3-fold dilutions, 10 points)). Abbreviation: GR: glucocorticoid receptor.

**Figure 4 cells-12-02291-f004:**
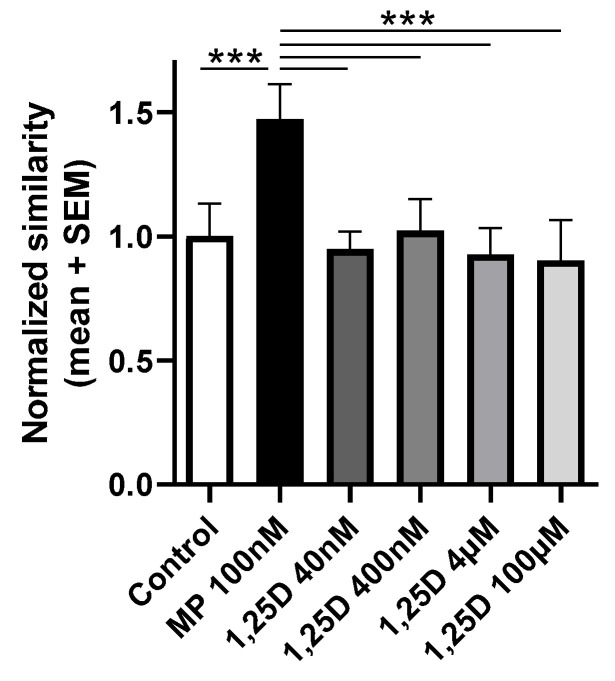
The ability of calcitriol to induce the nuclear translocation of GR was investigated. Hela ω3 cells expressing mCherry-H2B in the nucleus were incubated for 2 h with control, MP 100 nM, or different concentrations of calcitriol (40 nM, 400 nM, 4 µM, 100 µM). ImageStream system. Data are presented as the normalized similarity index between the nucleus and the GR inside the cell. Abbreviations: MP: methylprednisolone, SEM: standard error of the mean, 1,25D: calcitriol. Statistics were performed using a Repeated Measure One-way ANOVA followed by Newman Keuls Multiple Comparison test: *** < 0.001.

**Figure 5 cells-12-02291-f005:**
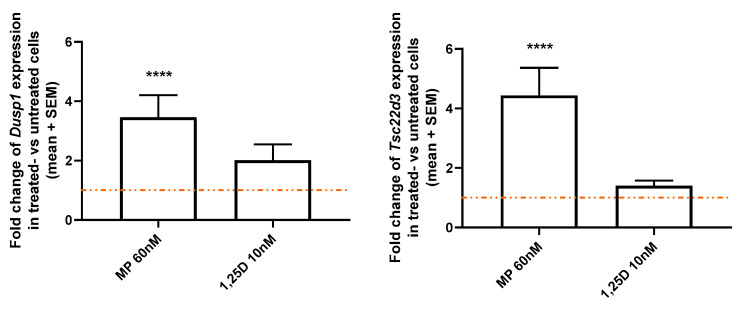
Gene expression of *Dusp1* and *Tsc22d3* was analyzed in stimulated splenic-derived CD3+ T cells of WT mice treated with control, MP 60 nM or calcitriol 10 nM (*n* = 5 in 1 to 3 replicates; stimulus conA 1.5 µg/mL; 3 h). Gene expression of treated cells was normalized to the one of untreated cells. RT-qPCR. Abbreviations: MP: methylprednisolone, SEM: standard error of the mean, 1,25D: calcitriol. Statistics: Friedman ANOVA test: **** < 0.0001.

**Figure 6 cells-12-02291-f006:**
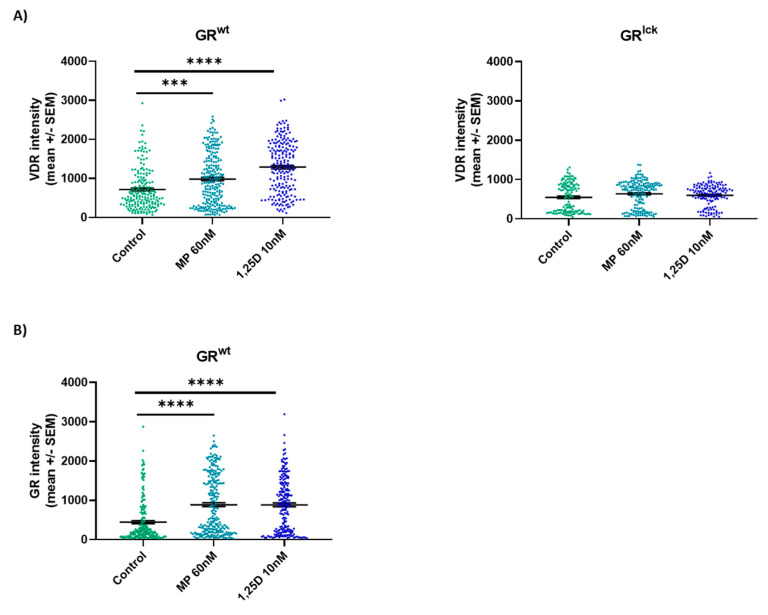
VDR and GR protein expression was analyzed in stimulated splenic-derived CD3+ T cells of WT and GR^lck^ mice (**A**) and of WT mice (**B**), respectively, treated with control, MP 60 nM or calcitriol 10 nM (*n* = 3; stimulus conA 1.5 µg/mL; 2 h). Confocal microscopy. Abbreviations: GR: glucocorticoid receptor, GR^lck^: T cell-specific GR-deficient mice, GR^wt^: WT mice, MP: methylprednisolone, SEM: standard error of the mean, VDR: vitamin D receptor, 1,25D: calcitriol. Statistics: Kruskal–Wallis test: *** < 0.001, **** < 0.0001.

**Figure 7 cells-12-02291-f007:**
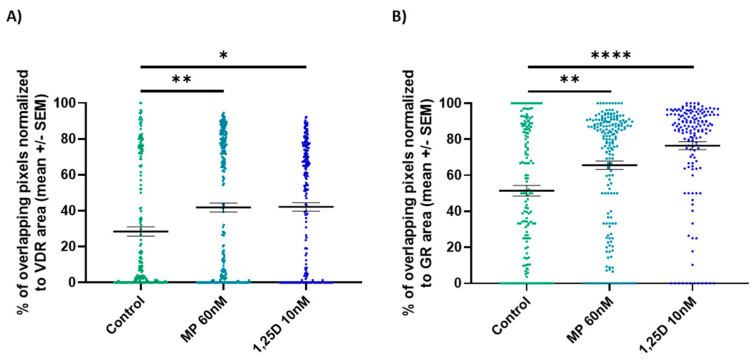
The co-localization of the GR and VDR (**A**,**B**) was analyzed in stimulated splenic-derived CD3+ T cells of WT mice treated with control, MP 60 nM or calcitriol 10 nM (*n* = 3; stimulus conA 1.5 µg/mL; 2 h). Confocal microscopy. Abbreviations: GR: glucocorticoid receptor, MP: methylprednisolone, SEM: standard error of the mean, VDR: vitamin D receptor, 1,25D: calcitriol. Statistics: Kruskal–Wallis test: * < 0.05, ** < 0.01, **** < 0.0001.

**Figure 8 cells-12-02291-f008:**
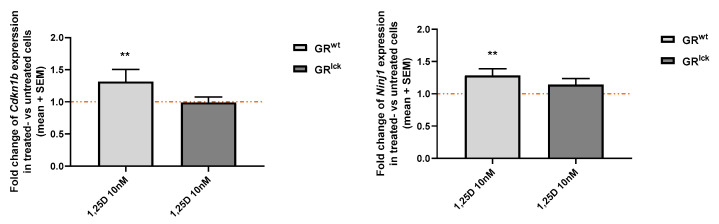
Gene expression of *Cdkn1b* and *Ninj1* was analyzed in stimulated splenic-derived CD3+ T cells of WT and GR^lck^ mice treated with control or calcitriol 10 nM (*n* = 5 in triplicates; stimulus conA 1.5 µg/mL; 3 h). Gene expression of treated cells was normalized to the one of untreated cells. RT-qPCR. Abbreviations: GR^wt^: WT mice, GR^lck^: T cell-specific GR-deficient, mice SEM: standard error of the mean, 1,25D: calcitriol. Statistics: Wilcoxon signed rank sum test: ** < 0.01.

## Data Availability

Data are available via the corresponding author upon reasonable request.

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
