# Peer review of "In Vivo and In Vitro Evidence for an Interplay between the Glucocorticoid Receptor and the Vitamin D Receptor Signaling"

_cells, 2023, doi:10.3390/cells12182291_

Round 1

Reviewer 1 Report

In this manuscript, the authors address the interplay between the glucocorticoid receptor and the vitamin D receptor in CD3+ T cells. The overall idea is interesting as previous work by the same authors reported the beneficial effects of combined treatments with Vitamin D and glucocorticoids in an experimental model of multiple sclerosis. However, there are several major drawbacks including that authors do not actually show the interaction between GR and VDR; and do not assess the effects of the combined treatment in the mechanistic experiments.

Specific points

Fig. 1. This figure is virtually identical to that shown by Hoepner et al., Acta Neuropathologica (2019) 138:443–456, which included the effects of combined treatment on the disease course. The only add of this figure is showing a dose response, which could be interesting if combined treatment was also shown. In any case, it is unclear why in GRlck T mice, the effects of calcitriol do not follow a dose-response curve (worsen disease score at 10 ng and 1 microg but no effects at 100 ng). Moreover, in the previous work, 10 ng of 1,25D had no effects while in this, the same dose worsens. These differences need to be explained, and state clearly the time when the disease scores is statistically changed (from day X). The color codes make difficult to follow up the graph.

Fig. 2. Saying that calcitriol induces T reg differentiation and CD3+ apoptosis in WT but not GRlck T mice is an overstatement. Despite statistical significance, the relative increases are marginal, in particular the numbers of apoptotic cells. One wonders about the biological relevance of such changes. Also, data from WT and GRlck T mice should be shown in one figure to be able to compare percentages of differentiation or apoptosis between genotypes. Again, combined treatment should be tested.

Fig. 4. The positive control for GR nuclear translocation –MDP- gave a very small increase, around 1.5-fold. That makes one wonder whether the technique is limiting to detect small increases of GR nuclear translocation upon 1,25D treatment. In Methods, authors state that cells were incubated with 10% GC-stripped FBS (Sigma). Is this equivalent to charcoal-stripped or DCC-serum? For how long were cells treated?

Other techniques could be used to confirm the lack of GR translocation such as using GRE-reporter assays; or better ChIP-qPCR to bona fide GRE sites. For example, for ligand-bound VDR and GR at the adjacent VDRE and GRE of MKP-1 gene promoter, as commented in the discussion (it would be also useful searching databases).

 Fig. 5. What motivated the choice of Dusp1 and Gilz as GR targets in the context of splenic T cells? Statistical analyses are a problem as standard deviation should be also shown in controls (even they are normalized to one; Figs. 5 and 7). It is hard to believe that at least Dusp1 is not upregulated by 1,25D (around 2-fold increase). Is it discarded that these genes are also regulated by vitamin D? Additional genes should be tested by RT-qPCR, and combined treatments included.

Fig. 7. As commented above, the reason for choosing Cdkn1b and Ninj as VDR targets in this context should be explained. In fact, Cdkn1b is also a GR-target. Again, the relative induction is below 1.5 fold, making hard to believe the overall differences between genotypes. Control value for WT is usually set as one (normalization; again, add SD); but it is important to show the relative values in GRlck T mice for comparison purposes. Additional genes –with biological relevance in the context of T cells- should be tested by RT-qPCR, and combined treatments included in the experiments.

Fig. S1. Given the Kd of vitamin D, what is the point of testing 100 microM dose?

Fig. S2. As VDR is constitutively nuclear, co-localization experiments do not really proof that both receptors go to the nucleus upon treatment.

As this paper addresses mechanistic aspects, one major point is demonstrating the interaction between VDR and GR. Authors should show the expression of VDR and GR in T cells from WT and GRlck T mice by immunoblotting, testing nuclear translocation by the respective ligands (and in combination). This will allow for quantitation of results. Also, the physical interaction between GR and VDR in T cells needs to be assessed by co-IP, PLA, etc

Discussion. Is it possible that calcitriol modifies the expression or activity of one common interactor such as MED14, resulting in increased GR-dependent action? This or similar possibilities would add to mechanistic insights

Do GRlck and control mice show different serum levels of vitamin D? As T cells are able to produce vitamin D, and also glucocorticoids, it would be very important to assess 1,25D levels and corticosterone in both WT and GR lck mice

Author Response

Dear Reviewer,

Thank you very much for having reviewed our paper. Please find in attachement a point-by-point reply.

Best regards

Reviewer 2 Report

The study aimed to explore the relationship between the glucocorticoid receptor (GR) and the vitamin D receptor (VDR) signaling. The authors found that GR plays a role in the vitamin D-induced effects in experimental autoimmune encephalomyelitis (EAE) using wild-type and T cell-specific GR-deficient mice. In addition, they discovered that vitamin D-enhanced T cell apoptosis and T regulatory cell differentiation are reduced in vitro in CD3+ T cells of GR-deficient but not WT mice. Although vitamin D does not signal directly via GR, it was observed that GR and VDR spatially co-localize after vitamin D treatment. The study suggests that a functional GR, specifically in T cells, is necessary for VDR to signal appropriately to facilitate the therapeutic effects of vitamin D.

However, there are a few issues that limit the enthusiasm of this reviewer:

1. As they have acknowledged several times in the manuscript, the calcitriol treatment efficacy in WT vs. GR-deficient mice (Fig 1) was reported by the authors in the past (Citation#23). Similarly, the lack of calcitriol-induced apoptosis of T cells in GR-deficient mice (Fig 2B) was reported before by the authors themselves (Citation#23, using GR-dim mice, mice with functional impairment of GR receptor dimerization). The authors have confirmed these data again here in this study which is promising; however, this diminishes the novelty and the actual original contribution of this paper.

2. This would have been okay if the authors had done more extensive work understanding and solidifying the molecular mechanism behind this purported relationship. However, the authors have stopped short of doing so. For example, the authors have shown that calcitriol doesn’t bind directly to GR and doesn’t induce nuclear translocation of the GR. But to explore why there is a phenotypic connection between the VDR and GR signaling, they only checked the expression of a couple of handpicked genes and concluded that there was no impact of calcitriol on the GR signaling.

They have taken a similar approach when investigating the extent of the impact of GR on VDR signaling after calcitriol treatment in T cells, which seriously undermines the contribution of this work.

The authors need to address these critical questions, which they attempted to investigate in this study, by employing a bulk-RNA sequencing approach in WT and GR-deficient T cells with or without calcitriol treatment.

3. It would be interesting to see whether similar GR signaling is also required in other aspects of the Treg cell functionalities, such as their suppressive function. The authors could shed light on this by performing a few simple in vitro experiments.

Author Response

(The authors gave the same response as above.)

Round 2

Reviewer 1 Report

The authors have addressed most issues, and new graphs in this revised version represent an improvement. Also, the effort of performing transcriptomic analyses is appreciated (though it was not specifically requested). However, one remaining concern is the lack of experimental evidences supporting one major point in the MS: “Mechanistically, VitD does not signal directly via the GR, as it does not bind to the GR, does not induce its nuclear translocation, and does not modulate GR-induced gene expression”. It is unfortunate that there are no RNA samples left to test few additional genes; therefore, authors need to state that data shown represent specific examples of (lack of) gene regulation by vit D instead a general mechanism. While the difficulty of establishing new methods in the lab is perfectly understandable, if a more general conclusion is to be reached, additional experiments are required. Both luciferase experiments and searching public data bases are feasible tools nowadays.

In Fig. 2, the observed 2% difference between treated and untreated cells has to be indicated in the text. Which evidences support “a constantly increased apoptosis of 2%”, or the actual loss of approximately half of the tissue/organ? Whether these are facts or speculations have also to be clearly stated.

Author Response

Dear Reviewer,

We thank you very much for reviewing the R1 version of our manuscript. Please find in attachement a point-by-point reply.

Best regards,

Reviewer 2 Report

The authors have, within their capabilities, reasonably tried to address the concerns raised.

Author Response

Dear Reviwer,

We thank you for taking into consideration the efforts we made to answer your comments.

Best regards,

Round 3

Reviewer 1 Report

Authors have addressed most questions